# SPATIO-TEMPORAL AND CROSS-SCALE INTERACTIONS IN HYDROCLIMATE VARIABILITY: A CASE-STUDY IN FRANCE

Manuel Fossa[1], Bastien Dieppois[2], Nicolas Massei[1], Matthieu Fournier[1], Benoit Laignel[1], and Jean-Philippe Vidal[3]

[1]  Normandie Univ, UNIROUEN, UNICAEN, CNRS, M2C, 76000 Rouen, France
[2]  Centre for Agroecology, Water and Resilience (CAWR), Coventry University, Coventry, UK
[3]  INRAE, UR Riverly, 5 rue de la Doua, CS 20244, 69625 Villeurbanne Cedex, France

**Correspondence:** Manuel Fossa (manuel.fossa1@univ-rouen.fr)

**Abstract.** Understanding how water resources vary in response to climate at different temporal and spatial scales is crucial to inform long-term management. Climate change impacts and induced trends may indeed be substantially modulated by low-frequency (multi-year) variations, whose strength varies in time and space, with large consequences on risk forecasting systems. In this study, we present a spatial classification of precipitation, temperature and discharge variability in France, based on a fuzzy clustering and wavelet spectra of 152 near natural watersheds between 1958 and 2008. We also explore phase-phase and phase-amplitude causal interactions between time scales of each homogeneous region. Three significant time scales of variability are found in precipitation, temperature and discharge: 1 year, 2-4 years and 5-8 years. The magnitude of these time scales of variability is however not constant over the different regions. For instance, Southern regions are markedly different from other regions, with much lower 5-8 years variability and much larger 2-4 years variability. Several temporal changes in precipitation, temperature and discharge variability are identified during the 1980s and 1990s. Notably, in the Southern regions of France, we note a decrease in annual temperature variability in the mid 1990s. Investigating cross-scale interactions, our study reveals causal and bi-directional relationships between higher and lower-frequency variability, which may feature interactions within the coupled land-ocean-atmosphere systems. Interestingly, however, even though time-frequency patterns (occurrence and timing of time scales of variability) were similar between regions, cross-scale interactions are far much complex, differ between regions, and are not systematically transferred

from climate (precipitation and temperature) to hydrological variability (discharge). Phase-amplitude interactions are indeed absent in discharge variability, although significant phase-amplitude interactions are found in precipitation and temperature. This suggests that watershed characteristics cancel the negative feedback systems found in precipitation and temperature. This study allows for a multi-time scale representation of hydro-climate variability in France, and provides unique insight into the complex non-linear dynamics of this variability, and its predictability.

# 1 Introduction

Hydroclimate variability represents the spatio-temporal evolution of hydrological (e.g. discharge, groundwater level) and climate variables (e.g. precipitation and temperature), which are directly impacting hydrological variability. Studying how hydrological variables react to climate variability and change is a major challenge for society, in particular for water resource management, flood and drought mitigation planning (IPCC, 2007, 2014, 2021). However, hydrological variability is expressed at multiple time scales (Labat, 2006; Schaefli et al., 2007; Massei et al., 2007, 2017), for which driving mechanisms remains poorly characterised and understood. As suggested in Blöschl et al. (2019), understanding the spatio-temporal scaling, i.e. how the general dynamics driving hydrological variability change at spatial and temporal scales, represents a major challenge toward improved prediction systems (Gentine et al., 2012). Understanding spatio-temporal scaling required to identify regions, i.e. the maximum spatial scale in which the dynamics remain unchanged despite its non-linearity, is critical (Hubert, 2001). Hydrological variability is by definition non-linear (Labat, 2000; Lavers et al., 2010; McGregor, 2017), as it results from complex interactions between atmospheric dynamics and catchment properties that may vary at different time scales (e.g. soil characteristics, water table, karstic systems, vegetation covers;(Gudmundsson et al., 2011; Sidibe et al., 2019)). Such interactions between processes at different time scales, i.e. cross-scale interactions (Paluš, 2014; Jajcay et al., 2018), have never been studied to further understand hydrological variability. It has also been shown that hydroclimate variability is inherently non-stationary, with time dependence of the mean and variance due to changes in the controlling factors (e.g. Coulibaly and Burn (2004); Labat (2006); Dieppois et al. (2013, 2016); Massei et al. (2017)).This results in difficulties in characterizing and predicting the hydrological variability at different spatio-temporal scales (Gentine et al., 2012; Blöschl et al., 2019).

While different time scales have been identified in hydrological variability (Coulibaly and Burn, 2004; Labat, 2006; Dieppois et al., 2013, 2016; Massei et al., 2017), very little has been done to: i) explore how spatially coherent are those time scales, ii) identify regions in which the statistical characteristics of all ranges of variability remain unchanged. Studying 231 stream gauges throughout the world, Labat (2006) highlighted different time scales of discharge variability over the different continents. At the regional scale,

Smith et al. (1998) established a clustering of 91 US stream gauges based on their global wavelet spectra, i.e. dominant time scales, and found five homogeneous regions. Similarly, Anctil and Coulibaly (2004) and Coulibaly and Burn (2004) established a clustering of Southern Québec and Canada streamflow, based on the timing of both the 2-3 and 3-6 year time scales. In Europe, Gudmundsson et al. (2011) identified different regions according to the magnitude of decadal discharge variability. In France, such a cluster-

ing, based on time-frequency patterns of discharge variability, as well as its relation to climate variability (e.g. precipitation and temperature), has not yet been explored. In addition, all studies mentioned above either isolated particular time scales of variability or averaged the variability across time scales (e.g. global wavelet spectra), which is equivalent to a linearization of the system (Hubert et al., 1989). These studies thus ignore potential feedback mechanisms, *e.g.* between soil moisture, precipitation, and temperature

(Materia et al., 2021; Ardilouze et al., 2020; Bellucci et al., 2015). In the presence of feedback mechanisms, interactions occur over different tim However, while studying cross-scale interactions have gained increasing interest in other fields, such as neurosciences (*e.g.* Onslow et al. (2014); Wang et al. (2014)), cross-scale interactions are poorly understood in climate and hydrological sciences. New strategies have recently been developed to facilitate such studies (Jajcay et al., 2018). Cross-scale interactions are however very relevant to hydroclimate studies, in particular

when searching for climate drivers (or predictors) of hydrological signals, as they will reveal climate time scales that are causality linked to each time scale of hydrological variability.

    In this study, we investigate the spatial homogeneity of hydroclimate variability in France, across time scales. We aim at identifying homogeneous regions according to specific time-frequency patterns? From the determination of homogeneous regions of hydroclimate variability, we will explore cross-scale interactions

that may result from feedback processes between catchment properties and hydroclimate variability.

    This study therefore have major implication for the comprehension of hydroclimate dynamics and their interactions with large-scale climate drivers, and catchment properties. In addition, as recently suggested in Scaife and Smith (2018), improved characterization of the different timescales of variability and their

interactions could help optimising ensemble-based hydrological forecasting systems, through identifying
climate ensemble members that better match the observed realisation.

The work is divided into the following sections. Data and methods are introduced in Section 2. In Section
3, we establish homogeneous regions for precipitation, temperature and discharge variability based on their
time-frequency patterns and then explore cross-scale interactions for each region of homogeneous variability
in precipitation, temperature and discharge. Finally, discussions of the main results and conclusions are
provided in Section 4.

## 2 Data and methodology

### 2.1 Hydrological and climate data

The data consist of precipitation, temperature and discharge time series located over 152 watersheds (Figure 1,2a,2b) Discharge time series were extracted from French Reference Hydrometric Newtork compiled
by Giuntoli et al. (2013b). This network of stations identifies near-natural watersheds (*i.e.* with negligible anthropogenic modifications) with long-term high-quality hydrometric data. According to Giuntoli et al. (2013a)
this subset of stations does not show abrupt changes and trends that could have resulted from anthropogenic
influence. The period 1968-2008 was chosen by Giuntoli et al. (2013b) as being the best trade-off in terms
of data availability over the different regions. Here, this database was further subset to 152 watersheds in
order to select complete monthly timeseries (i.e. without missing values), only (Figure 1). Precipitation and
temperature data have been estimated from the 8 km-grid Safran surface reanalysis dataset (Vidal et al.,
2010), and has been subset to a common period (1968-2008). For this study, precipitation and temperature
have been averaged over each watershed area (Caillouet et al., 2017). Each station is thus representative of
one watershed.

### 2.2 Methods

The methodology described below, and summarized on the workflow (Figure 2), is applied to precipitation,
temperature and discharge data sets.

### 2.2.1 Continuous wavelet transforms

For each of the 152 watersheds continuous Wavelet analysis is used to identify at which time scales and time locations the amplitude of variability (*i.e.* local variance) is the strongest (Figure 2c, (Torrence and Compo, 1998)). Here we employ interchangeably the words "time scale" and "frequency", though frequency implies a periodic variability, which is not a necessary condition in continuous wavelet analysis. For any finite energy signal $x$, it is possible to obtain a time-frequency representation by projecting the time series on a function called the mother wavelet, which quantifies the amplitude of the time series variability at a given time scale and time location. This mother wavelet can be translated in time, to quantify the variability at precise time locations, but also scaled so that variability at different time scales can be quantified as well (Torrence and Compo, 1998; Grinsted et al., 2004). A mother wavelet at a time scale $a$ and time location $b$ is called a daughter wavelet. Daughter wavelets are calculated as:

$$\psi_{a,b} = \frac{1}{\sqrt{a}} \psi(\frac{t-b}{a}) \tag{1}$$

The left hand side (LHS) term is the daughter wavelet of scale $a$ and time translation $b$ at time $t$. For the sake of simplicity, we will refer to $b$ as the time location. The first right hand side (RHS) term is the scaling of the mother wavelet $\psi$ and the last one is the time translation. The projection of the signal onto each scale $a$ of the form:

$$WT_{\psi}[x](a,b) = \langle x, \psi_{a,b} \rangle = \int_{R} x(t)\psi_{a,b}(t)dt \tag{2}$$

LHS term contains the wavelet coefficients $WT$, *i.e.* how large is the amplitude of variability at the time scale $a$ and time location $b$. If the mother wavelet (and hence the daughter wavelets as well) is complex, wavelet coefficients are complex as well , and both the amplitude and instantaneous phase of the time series can be computed around time location $b$ and time scale $a$ . Wavelet coefficients represent the inner product of the signal, daughter wavelet of scale $a$ and time location $b$ (Centre). The norm of their square is called the wavelet power and represents the amplitude of the oscillation of signal $x$ at scale $a$ and centred on time location $b$. As it is impossible to capture the best resolution in both time scale and time location simultaneously, here, we used a Morlet mother wavelet (order 6), which offers a good trade-off between detection of scales and localisation of the oscillations in time (Torrence and Compo, 1998). Visualization of a continuous

wavelet transform is called a scalogram. Figure 2c shows a collection of scalograms, with time location on

the horizontal axis, and time scale on the vertical axis. Yellow colors show the time scales and time locations, when the amplitude of the time series' variability is maximum. A major advantage of continuous wavelet transform compared to other signal analysis methods, such as the Fourier transform, is that wavelet analysis takes non-stationarity into account. Non-stationarity is the time location dependence of both the mean and variance of a time series.

Because the daughter wavelet translates and scales up, overlap in time and frequency can occur, and wavelet coefficients can be overestimated, requiring statistical significance tests (Torrence and Compo, 1998). This redundancy may give rise to peaks in the wavelet coefficients (meaning, high variability detected) even in the case of a random noise (Ge, 2007). Torrence and Compo (1998) used Monte-Carlo simulations to assess the statistical significance of the co

For the reminder of this study, the terms "intra-seasonal","annual", and "inter-annual" refer to variations at "<1yr", "1yr", "2-4yr" and "5-8yr" time scales, respectively.

### 2.2.2 Image Euclidean Distance Clustering

After each watershed wavelet spectrum is computed, we estimate the similarity between them, *i.e.* how sim-

ilar is the variability, for given scales and time locations, among all wavelet spectra (Figure 2d). Similarities between wavelet spectra are estimated from the entire wavelet spectrum, and not only on statistically significant signals, to guarantee more consistent comparison between spectra. Distances between two-dimensional data, such as wavelet spectra, are estimated using Euclidean distance between pairwise points (pED; i.e computing $f_2(x_i, y_i) - f_1(x_i, y_i)$). However, such a procedure has no neighborhood notion, making it impossible

to account for globally similar shapes (Wang et al. (2005)). To avoid this issue, we used the Image Euclidean distance calculation method (hereinafter IEDC) developed by Wang et al. (2005). The IEDC method modifies the pED equation in two ways (Wang et al., 2005): i) the distance between pixel values is computed not only pairwise, but for all indices; ii) a Gaussian filter, function of the spatial distance between pixels, is applied. The Gaussian filter then applies less weight to the computed distance between very close and far

apart pixels, while emphasizing on medium spaced ones (Wang et al., 2005).

### 2.2.3 Fuzzy clustering

Fuzzy clustering has then been used to cluster the different watershed based on their similarities (Figure 2e). Fuzzy clustering is a soft clustering method (Dunn, 1973). While soft clustering spreads membership over all clusters with varying probability, hard clustering attributes each station one and only one cluster membership. Soft clustering is therefore better-suited when the spatial variability, originating from different stations' hydroclimate characteristics, is smooth. For instance, precipitation and temperature patterns are unlikely to change suddenly from one station to a neighboring one, and in turn, be markedly different from the next neighbor (Moron et al., 2007; Hannaford et al., 2009; Rahiz and New, 2012). As such, several stations tend to show transitional or hybrid patterns, and can potentially be member of different clusters, limiting the robustness of hard clustering procedure (Liu and Graham, 2018).

Fuzzy clustering performance is determined by the ability of the algorithms to recognize hybrid stations (i.e. stations incorporating multiple features from different patterns observed in other coherent regions), while allowing for a clear determination of the membership of stations with unique features (Kaufman and Rousseeuw, 1990). Here, we used the FANNY algorithm (Kaufman and Rousseeuw, 1990), which has been shown to be flexible, and to offer the possibility to adapt the clustering to the data, with optimal performance (Liu and Graham, 2018). In addition, rather than setting the number of clusters arbitrarily, we used an estimation of the optimum number of clusters by first computing a hard clustering method: the consensus clustering (Monti et al., 2003). Thus, the number of clusters providing the best stability (i.e. the minimal changes of membership when adding new individuals) is considered optimal as recommended in Şenbabaoğlu et al. (2014). The different clusters' memberships are then mapped to discuss the spatial coherence of each hydroclimate variable.

### 2.2.4 Cross-scale interactions

For each variable and each cluster, cross-scale interactions are explored (Figure 2f). Cross-scale interactions refer to phase-phase and phase-amplitude couplings between time scales of a given time series (Paluš, 2014; Scheffer-Teixeira and Tort, 2016). Here, coupling means that the state (either phase or amplitude) of a signal $y$ is dependent on the state of a signal $x$, and describes causal relationship (Granger, 1969; Pikovsky et al., 2001), which refers to information transfer from a time scale of a signal to another.

Figure 3 describes the necessary setting and characteristics of cross-scale interactions. A variable $f$ measures the dynamics of a a system (e.g. precipitation or temperature variability). This system is modelled as a coupling of two components $X$ and $Y$. The components interact with each other in a perturbation-dampening ($X$, $Y$, respectively), so that $f(t) = X(..,t) - Y(..,t)$ (Figure 3a). The interactions between the components occur through the connections $C_{XY}$ and $C_{YX}$, with a given strength (here $C_{..} = 2$), and this perturbation-dampening interaction forms a negative feedback (*i.e.* increase in $X$ activity triggers $Y$ activity, dampening $X$ activity which in turn lowers $X$ activity, thus lowering $Y$ activity, allowing $X$ activity to increase again, and so on (Figure 3a).). The connection $C_{XX}$ enables $X$ to grow first before $Y$ dampens it. This connection $C_{XX}$ forms a positive feedback. *i.e.* increases in $X$ activity will be more severe as $X$ activity is high. Both $X$ and $Y$ receive inputs $\phi_X$, $\phi_Y$ from driving processes (*e.g.* moisture advection, convective processes) (Figure 3a). Depending on both the mean and time scales of $\phi_X$ and $\phi_Y$, the strength of $C_{XX}$, $C_{XY}$ and $C_{YX}$, $X$ and $Y$ may show coupled behaviors. For instance, in Figure 3b, every fourth ridges of $Y_{PP}(t)$ is synchronized with a ridge of $X(t)$ (Figure 3b, top and middle panels), thus forming a phase-phase interaction. The direction of the interactions depends on the inputs $\phi_X$, $\phi_Y$, and the connections $C_{XX}$, $C_{XY}$ and $C_{YX}$. $f_{PP}(t)$ is the difference between $X(t)$ and $Y_{PP}$ (Figure 3b, bottom panel). Because the interaction between $X$ and $Y$ depends on both inputs and connections, interactions may lead to a cross-scale relationship only for certain values of $X$ or $Y$ (Figure 3c). Thus, depending on the phase of either $X$ or $Y$, the amplitude of the driven component may increase/decrease when the cross-scale interaction takes place, and return to normal when it is out of phase compared to the driving component. This describes a phase-amplitude interaction. In Figure 3b, $Y_{PA}$ amplitude decreases when $X$ is at its maximum (*i.e.* when its phase is a ridge; Figure 3c, top and middle panels). Similarly to the phase-phase interaction, $f_{PA}(t)$ is the difference between $X(t)$ and $Y_{PA}(t)$ (Figure 3c, bottom panel). Because phase amplitude are very dependent on inputs $\phi_X$ and $\phi_Y$, connections between spatially distant physical processes are likely to give rise to phase-amplitude interactions (Nandi et al., 2019). In summary, there are three elements needed for cross-scale interactions between multiple, coupled processes, to arise: i) an oscillating forcing $\phi_X$ must drive $X$. An additional forcing $\phi_Y$ on $Y$ may also be present; ii) $X$ must have positive feedback on itself so that it grows faster than $Y$; iii) $Y$ must show dampening effect on $X$ ($XY$ or $YX$ negative feedback). The presence and characteristics of cross-scale interactions depend on the strength and frequencies of $\phi_X, \phi_Y$, intrinsic frequencies of $X, Y$ and coupling strengths $C_{XY}, C_{YX}, C_{XX}, C_{YY}$ (Figure 3).Thus, detection of cross-scale interactions in time series is an

indication of the presence of all those characteristics in the hydroclimate system, *e.g.* precipitation-land processes, which helps in investigating potential processes at play.

The balance between $X$ and $Y$ determines if the feedback is either positive or negative (Peters et al., 2007).

Note that cross-scale interactions can occur from large-scale to small-scale processes, and vice-versa. For instance atmospheric circulation at seasonal time scales, influences inter-annual and decadal time scales, which in turn influence seasonal variations (Hannachi et al., 2017).

Following Paluš (2014) and Jajcay et al. (2018), we choose the conditional mutual information (CMI) surrogates method, combined with wavelet transforms. First, using a Morlet mother wavelet, the instantaneous

phase and amplitude at time $t$ and scale $s$ of the signal are obtained. Next, the conditional mutual information, $I(\phi_x(t); \phi_y(t+\tau) - \phi_y(t)|\phi_y(t))$ for the phase and $I(\phi_x(t); A_y(t+\tau)|A_y(t)), A_y(t-\eta)), A_y(t-2\eta))$ for the amplitude is computed. In the case of phase-phase relationships, the CMI measures how much the present phase of $x$ contains information about the future phase of $y$ knowing the present value of $y$. Phase-phase interactions can be uni- or bi-directional. It is possible for a single time scale to drive another, which in

turn, drives back the original one, describing feedback interactions. For phase-amplitude relationships, CMI measures how much the present phase of $x$ contains information of the future amplitude of $y$ knowing the present and past values of $y$. The statistical significance of the CMI measure is assessed using 5000 phase-randomized surrogates, having the same Fourier spectrum, mean and standard deviation as the original time series, as in Ebisuzaki (1997). Paluš (2014) has shown that this number of surrogates is ideal for statistical

significance, in the context of hydroclimate time series. The computational cost is however high, with approximately one week of computing for a time series of 50 years, on a 32-core xeon computer. The present computations were done on the Myria cluster, hosted by the Centre Régional Informatique et d'Applications Numérique de Normandie (www.criann.fr).

## 3   Spatio-temporal clustering of hydrological variability

The wavelet transform corresponding to each watershed's monthly time series have been computed and all 152 watersheds' wavelet transforms have been checked for similarities using IEDC fuzzy clustering to identify and characterize homogeneous regions of hydroclimate variability over France. Once the homogeneous regions have been identified, an average time series for each region was computed. The global wavelet spec-

trum of this time series quantified the total variance expressed at each time scale, while its wavelet spectrum
characterized how this variance is distributed in the time (location) and frequency (scale) domains. In addition, so as to focus on inter-annual time scales, we computed the wavelet spectrum of the time series filtered
at the annual time step. Cross-scale interactions were then investigated for each homogeneous region.

## 3.1 Precipitation

### 3.1.1 Time-frequency patterns

Seven regions with homogeneous time-frequency patterns are identified (Figure 4a): North-western (green),
North-eastern (blue), Centre-North (red), Centre-western (pink), Centre-eastern (black), South-western (yellow) and South-eastern (dark green). Figure 4a shows that all watersheds converge toward singular clusters,
meaning that all regions are highly coherent (*i.e.* pie charts in Figure 4a show one dominant color).

In all regions, precipitation is varying at different time scales, ranging from intra-seasonal to inter-annual
scales (i.e. 2-8 years; Figure 4b). South-western and -eastern are dominated by annual (1yr) variability,
while their inter-annual variability (2-8yr) is low and reversely for other regions. In addition, statistically
significant areas in continuous wavelet spectra however show that those time scales of variability are nonstationary (Figure 4c), with temporal changes in terms of amplitude discriminating the different regions. For
instance, South-western regions are characterized by quasi-continuous significant annual variability until
the late 1980s, while other watersheds show sparsely significant annual variability (Figure 4c). Similarly,
although there is significant inter-annual variability in all watersheds from the late 1980s, during this period,
South-western and -eastern regions do not show significant inter-annual variations (Figure 4c). After removing the <=1yr time scales (*i.e.*the seasonal cycle), focusing on inter-annual time scales, significant variability
at 2yr (South-western) and 5yr (South-western and South-eastern) time scales emerge for the Southern regions. The largest variations (*i.e.* coloured circles) occur over shorter periods of time than in other regions
(Figure 5b).

In summary, different regions with coherent precipitation variability are identified, and are characterised
by three time scales of variability: intra-seasonal, annual and inter-annual. The amplitude of those time scales
of variability however differs in time and over the French territory. Mediterranean regions (South-western
and -eastern) have comparatively weaker inter-annual variability as compared to annual time scales. The

differences between regions are both dependent on the local expression of the climate forcing and watershed characteristics. Because those physical processes are in interaction, studying cross-scale interactions in precipitation brings more insight on the dynamics behind the spectral characteristics of each region (Boé, 2013; Materia et al., 2021; Bellucci et al., 2015; Ardilouze et al., 2020).

### 3.1.2 Cross-scale interactions

Figure 6 shows cross-scale interactions for each cluster of precipitation variability (cf. Figure 4).

North-eastern, South-eastern, North-centre, North-western and Centre-eastern regions all show the phase of a 5-8yr variability driving the variability of smaller time scales (Figure 6a, blue, dark green, red, green, black, lower half of the graph). This cross-scale interactions is however more pronounced in North-eastern and South-eastern regions (Figure 6a). Similarly, eastern regions exclusively show $5 - 8yr \rightarrow 2 - 4yr$ interactions, while other regions show self-interacting 5-8yr variability (Figure 6a). The upper half of the graph, which refers to higher-frequency driving lower-frequency variability, is populated by North-centre, South-eastern, North-western and North-eastern regions (Figure 6a, red, dark green, green and blue). South-eastern shows cascade phase-phase interactions, *i.e.* $2 - 3yr \rightarrow 5 - 4yr \rightarrow 6 - 5$ (Figure 6a, dark green). In addition, both South-eastern and North-western regions show mirror interactions with their lower half counterparts, *e.g.*, $5 - 6yr \leftrightarrow 4 - 5yr$ (Figure 6a, dark green, green mirror patches about the diagonal). We also note that phase-phase interactions are very weak over the South-western regions, and absent in the centre-western regions.

Phase-amplitude interactions are presented in Figure 6b. The lower half of the graph, which refers to lower-frequency driving higher-frequency variability, shows $5 - 8yr \rightarrow 2 - 4yr$ interactions for western and North-centre regions (Figure 6b, pink, yellow, green, red). Centre-eastern regions are also showing lower-frequency variability driving higher-frequency variability, but between $8yr \rightarrow 6yr$ variability (Figure 6b). Notably, the North-western region is the only one with cross-scale interactions driving the annual cycle (Figure 6a, green). In the upper half of the graph, which refers to higher-frequency driving lower-frequency variability, we only find North-centre and North-eastern regions, showing $2 - 4 \rightarrow 4yr$ and $3 - 4yr \rightarrow 7 - 8yr$ phase-amplitude interactions (Figure 6b, blue, red). Note that North-centre, North-eastern and Centre-eastern regions show phase-amplitude and phase-phase interactions at very similar time scales (Figure 6a,b, red, blue, black), while time scales of phase-amplitude and phase-phase interactions do not match in Centre-

western and North-western and South-western regions (Figure 6a, b, pink, green, yellow). Regions to the
East thus appear to have both phase-phase and phase-amplitude interactions at the same time scales, while
western regions are more characterized by phase-amplitude interactions.

The precipitation cross-scale interactions can be of different forms: phase-phase, phase-amplitude, uni-
or bi-directional, from lower to higher time scales and vice versa. The presence of cross-scale interactions
seems to be tied to specific spatial locations, suggesting different internal dynamics, over the different regions
of homogeneous precipitation variability. Interestingly, cross-scale interactions tend to converge toward spe-
cific time scales, notably 2-4yr and 5-8yr, which were linked to ocean-atmosphere variability, such as the
North Atlantic Oscillation, in previous hydroclimate studies over France (Feliks et al., 2011; Fritier et al.,
2012; Dieppois et al., 2016; Massei et al., 2017). In addition, the presence of mirror interactions also indicate
strong bidirectional negative feedback.

## 3.2 Temperature

### 3.2.1 Time-frequency patterns

In temperature, nine regions with homogeneous time-frequency patterns are identified (Figure 7a): North-
western-high (pink), North-western-low (black), North-eastern (blue), Centre-eastern (red), Centre-western
(green), South-eastern-high (yellow), South-eastern-low (brown), South-western-high (dark green) and South-
western-low (purple). Fuzzy clustering shows that watersheds typically converge toward singular clusters,
defining highly coherent regions (Figure 7a). This is however not true for the Centre-western region, which
is characterized by a mix of spectral characteristic defining other regions (cf. red, green, black, yellow and
purple pie charts, Figure 7a).

Using monthly data, temperature is primarily varying on an annual time scale, with very similar ampli-
tudes for all regions (Figure 7b-c).Since the dominant annual variability masks the other time scales, we use
the annual time step to study inter-annual variability (Figure 8a-b). Focusing on this inter-annual variability,
significant temperature variations indeed emerge at 2-4yr and 5-8yr time scales, and show different timings
and amplitudes over the different regions (Figure 8a-b). All regions show 5-8yr variability, but, compared
to northern regions, southern regions show significantly stronger variations on 2-4yr time scale (Figure 8a).

Similarly, while stronger 2-4yr variability occur in the 1980s and 1990s in the South-western-low region, other regions show significant 2-4yr variability from the 2000s, only (brown, Figure 8b).

### 3.2.2 cross-scale interactions

Figure 9 shows cross-scale interactions for each cluster of temperature variability identified in Figure 8a.

For temperature, phase-phase interactions are mostly concentrated in the upper half of the graph, which
refers to higher-frequency modulating lower-frequency (Figure 9a). Notably, a $2-6yr \rightarrow 6-8yr$ phase-phase interaction appears more pronounced over Northern regions (Figure 9a, blue, red, pink, black). The Centre-western region shows similar phase-phase interactions, but at $1-3yr \rightarrow 4-6yr$ time scales (Figure 9a, green). In the lower half of the graph, which refers to lower-frequency modulating higher-frequency, interactions are found at very similar timescales, but at slightly higher frequency, for all regions (e.g.
, $2-5yr \rightarrow 1-4yr$ variability,Figure 9a). Temperature in the South-western-low region, however, show slightly different characteristics with phase-phase interactions between lower- and higher-frequency occurring between $7-8yr \rightarrow 3-4yr$ and $7-8yr \rightarrow 3-4yr$ variability (Figure 9a, purple).

Temperature phase-amplitude interactions are mostly acting on the 3-4yr time scale for all regions (Figure 9b). In particular, in temperature, more pronounced phase-amplitude interactions are found over the South-
western-low region (Figure 9b, purple), consistently with previous studies on phase-amplitude interactions in European temperature (Palus, 2014; Jajcay et al., 2016). Over South-western regions, temperature, however, shows both $3-8yr \rightarrow 3-4yr$ and $2-4yr \rightarrow 4-7yr$ phase-amplitude interactions (Figure 9b, brown, purple). Furthermore, it should be noted that temperature variability interactions occur between very similar time scales over a number of regions (Figure 9b, pink, red, yellow, purple). According to Paluš (2014), interactions
between very similar time scales, or the same time scales, can only occur if, at least, two processes are present.

As for precipitation, in temperature, phase-phase and phase-amplitude cross-scale interactions are region-dependent, and can be uni- or bi-lateral. However, in temperature, most phase-phase interactions occur from higher- to lower-frequency variability, while phase-amplitude interactions tend to occur from lower-
to higher-frequency variability. Similarly, while time scales of variability that are involved for phase-phase and phase-amplitude interactions are very similar in precipitation, they differ largely in temperature (Figure 9b). This suggests that, in temperature, the processes driving phase-phase and phase-amplitude cross-scale

interactions are different. It also suggests that the processes driving cross-scale interactions are different in temperature and in precipitation..

## 3.3 Discharge

### 3.3.1 Time-frequency patterns

Six regions with homogeneous time-frequency patterns are identified in discharge (Figure 10a): North-western (black), North-eastern (blue), North-Centre (red), Centre-western (green), South-eastern (yellow) and South-western (pink). However, several watersheds, especially in the South, show memberships to multiple regions, suggesting lower spatial coherence in discharge than in precipitation and temperature. Lower spatial coherence, however, could mostly be explained by: i) mixing of solid and liquid precipitation in driving discharge variability in the Alps; and ii) the local heterogeneity of precipitation due to convective dynamics in the Pyrenees (Gottardi et al., 2008; Büntgen et al., 2008; Hermida et al., 2015) . Nevertheless, the number of significant homogeneous regions is lower in discharge than in precipitation and temperature, and northern regions are particularly coherent.

Using monthly data, discharge is mainly varying on annual time scales, as determined through the global wavelet spectra (Figure 10b). In addition, unlike other regions, South-eastern watershes shows significant intra-seasonal variability (Figure 10b). Continuous wavelet spectra show that both annual and intra-seasonal variability can be non-stationary, with temporal changes in terms of amplitude discriminating the different homogeneous regions of discharge variability (Figure 10c). For instance, annual variability is only significant for specific periods in the South-eastern watersheds, while other regions show quasi-continuous significant annual variability (Figure 10c). Similarly, in the South-eastern region, intra-seasonal discharge variability sparsely appears significant from the 1980s, while they are absent in other regions (Figure 10b).

After removing the seasonality, focusing on inter-annual variability, North-eastern watersheds stand out as having continuous significant inter-annual variability throughout the time series, with 4-5yr and 5-8yr variability before and after the 1990s, respectively (Figure 11b). South-eastern and -western regions also stand out, as they show 2-4yr variability in the mid-1970s and 2000s (Figure 11b, yellow, pink). In addition, South-eastern regions do not show significant variability in discharge at time scales greater than 4yr (Figure 11a-b).

70   Different coherent regions are thus identified for discharge variability. In addition, these homogeneous regions correspond well with regions identified in precipitation and temperature variability. As in precipitation and temperature, those regions seem strongly impacted constrainedperature, southern regions, which may appear more complex in term of climate and its link to land-surface processes, appear much less spatially coherent in discharge.

75   ### 3.3.2   cross-scale interactions

An important question concerning discharge cross-scale interactions is whether interactions found in either precipitation or temperature are also present in discharge. Phase-phase interactions that were found in precipitation are also identified in discharge, in particular over the North-eastern, South-eastern and North-western regions (blue, yellow and black; Figure 6a, Figure 12a). Phase-phase interactions that were identified in 80   temperature are much less evident (Figure 9a, Figure 12a). It should also be noted that many small patches, describing phase-phase interactions in precipitation and temperature, are systematically not transferred to discharge variability (Figures 6a, 9a, 12a). Instead, discharge variability seems to exclusively preserve large patches of phase-phase interactions (Figures 6a, 9a, 12a), suggesting that catchment properties are modulating the climatic signals (i.e. precipitation and temperature). Such filtering of climate signals is even more 85   pronounced in certain regions, such the North-centre, where phase-phase interactions are absent in discharge (Figure 12a), but were identified in precipitation and temperature (Figure 6a, 9a).

More importantly, there is no phase-amplitude interaction in discharge (Figure 12b). This points out that watershed properties modulate the interacting processes in precipitation and temperature.Because our data set is mostly composed of low groundwater support, those modulations are unlikely to result from the water 90   table, especially as phase-phase interactions are inherited from precipitation. In addition, further analysis on Paris' Austerlitz gauging station, which includes very large groundwater support, reveals the same absence of phase-amplitude interaction in discharge (not shown, Flipo et al. (2020)). Possible explanations include the frequency partitioning of watershed compartments or integration process along the river network breaks any spatial connection and thus smooths out and flattens phase-amplitude interactions (Schuite et al., 2019) 95   Cross-scale interactions are only of phase-phase nature in discharge. All phase-phase interactions in discharge seem to be primarily related to precipitation, even though the strong correlations between rainfall and temperature makes it difficult to detect. However, differences between regions of homogeneous discharge

variability are very similar to those detected in precipitation. Further work is however needed to understand why phase-amplitude cross-interactions are absent in discharge variability. Catchment properties appear to involve positive rather than negative feedback, thus resulting in a loss of phase-amplitude interactions.

## 4    Discussions and Conclusions

### 4.1    Spatial variability of homogeneous hydroclimate variability in France

As recommended by Blöschl et al. (2019), characterizing the different scales of spatial and temporal variability, as well as their interactions, remains one of the most important challenges in hydrology. In this study, we unravelled homogeneous regions of hydroclimate variability in France, accounting for non-stationarity and non-linearity, bringing additional information over previous, regime-based, classifications in France or elsewhere (Champeaux and Tamburini, 1996; Bower and Hannah, 2002; Sauquet et al., 2008; Snelder et al., 2009; Joly et al., 2010; Gudmundsson et al., 2011). This was achieved through a clustering analysis based on time-frequency patterns of precipitation, temperature and discharge variability over 152 watersheds. We then studied the spatio-temporal characteristics of each homogeneous region, including characteristic time scales of hydroclimate variability (i.e. precipitation, temperature and discharge) and cross-scale interactions.

Our study reveals different coherent regions of precipitation, temperature and discharge variability. Yet, some watersheds are characterized by a mix of spectral characteristics from surrounding regions, or regions with the same topographic characteristics. Those coherent regions are homogeneously distributed over France in precipitation and discharge, but show larger discrepancies in term of spatial extension in temperature. According to previous clustering of hydroclimate variability over France, Northern regions are more homogeneous than what was found here (Champeaux and Tamburini, 1996; Sauquet et al., 2008; Snelder et al., 2009; Joly et al., 2010), and show lower spatial coherence. In particular, here, we demonstrate that both the amplitude and timings of the different time scales of hydroclimate variability differentiate the regions, highlighting the need for accounting for non-stationary behaviours in global to regional hydroclimate study. Overall, hydroclimate variability displays intra-seasonal (<1yr), annual ( 1yr) and inter-annual (2-4yr and 5-8yr) time scales. Our results, which were focused on the French territory, are therefore consistent with time scales of variability identified over the world major rivers (Labat, 2006).

The time scales identified in this study have been shown to be important in climate processes, such as the

North Atlantic Oscillation, or the Gulf Stream front (Massei et al., 2007; Feliks et al., 2010; O'Reilly et al., 2017). Their interactions with watershed characteristics likely leads to their local expression with local processes, playing an important role in feedback mechanisms dampening or enhancing how the climate variability is expressed at the local scale(Haslinger et al., 2021; Materia et al., 2021; Bellucci et al., 2015).

## 4.2   cross-scale interactions

Feedback mechanisms can occur between any physical processes of the hydroclimate system, and identifying or attributing the nature of these processes is an intractable issue using observational data. Nevertheless, we can use the mandatory conditions for cross-scale interactions to arise, to discuss the processes that are potentially at play (Figure 3). In precipitation, cross-scale interactions involve lower-frequency time scales driving higher-frequency time scales. North Atlantic climate variability encompasses various processes, such

as North Atlantic Oscillation or sea surface temperature anomalies, that drives climate variability (Feliks et al., 2010; O'Reilly et al., 2016). Thus, moisture advection from the North Atlantic area could potentially act as a positive feedback forcing. Moisture advection has indeed been shown to impact western Europe precipitation, especially in wintertime (Sun et al., 2020; O'Reilly et al., 2017). Zonal moisture advection is only forcing precipitation variability when the region is not affected by blocking weather regimes

(Haslinger et al., 2019, 2021). Furthermore, vegetation, temperature and soil moisture, which are themselves interacting with each other, can act as a dampening forcing, dampening the precipitation. The precipitation-temperature, precipitation-soil moisture and precipitation-vegetation feedback have indeed been shown to reach a negative sign depending on prior state of the soil (Liu et al., 2006; Berg et al., 2015; Liu et al., 2006). However, the sign of temperature-soil moisture-vegetation feedback on precipitation have been shown to

be spatially dependent at the global scale. For instance, while temperature and soil-moisture have large effects in Western Europe, vegetation feedback is stronger and mostly of positive sign in Northern Europe (Woodward et al., 1998; Liu et al., 2006; Yang et al., 2018).In our results, South-eastern region shows inter-annual phase-phase interactions (Figure 6a) in contradiction with recent literature: for instance, in the Mediterranean Region, Ardilouze et al. (2020) found no negative soil-moisture precipitation feedback for

inter-annual scales, however, the authors used two climate models to simulate soil moisture sensitivity to precipitation forcing, and note that this variability is much larger in in century long reanalysis, such as

NOAA's 20CR. For other regions, inter-annual negative soil-moisture feedback was found by Boé (2013), while Sejas et al. (2014b) found negative ocean-land temperature differences precipitation feedback. Similar results were found in Bellucci et al. (2015) where interactions between compartments of the atmospheric circulation at intraseasonal time scale were found to produce significant interannual variations..

-annual temperature variability is tied to both soil state and atmospheric circulation, but that relation is location dependent. Large scale patterns, such as the North Atlantic Oscillation are shown to be source of both inter-annual precipitation and temperature variability, especially during winter-time, including for Southwestern France (Pepin and Kidd, 2006; O'Reilly et al., 2016). At more local scales, sea surface temperature anomalies has have been shown to interact with near-surface air temperature through sea-land heat exchanges regions(Lambert et al., 2011; Sejas et al., 2014a; Zveryaev, 2015). Soil moisture and evapotranspiration demand can enhance or dampen near-surface temperature variability (Miralles et al., 2012; Materia et al., 2021). Here, in temperature, phase-phase interactions are particularly interesting because they arise from higher frequency time scales driving lower frequency time scales (Figure 9a). As shown by Peters et al. (2004, 2007), higher frequency processes can spread to lower frequency ones by the means of intermediate time scales processes. High-frequency soil-moisture enhancing lower frequency large scale circulation may explain temperature cross scale interactions.

Regarding cross-scale interactions in discharge variability, the absence of phase-amplitude was particularly interesting. As our discharge data set is mostly composed of low groundwater support, the absence of phase-amplitude interactions is unlikely to result from the water table, especially as phase-phase interactions are inherited from precipitation. To test this hypothesis, we computed cross-scale interactions on the gauging station at Paris Austerlitz, which was not included in our original dataset, as it shows large groundwater support and anthropogenic influence. Results at Paris Austerlitz are consistent with other regions, and do not show any phase-amplitude interactions (not shown). As it has been shown that spatial heterogeneity (in the variable dynamics) favors cross-scale interactions, one possible explanation is that converging of runoff into the main drain cancels that spatial heterogeneity and thus phase-amplitude variability (Peters et al., 2007).

In this study, we interpreted cross-scale interactions based on the mandatory structure for such interactions to arise, the identified interacting time scales, comparison of cross-scale interactions in both precipitation, temperature and discharge. Dedicated studies are needed to explore in depth the drivers of those interac-

tions, as feedback mechanisms are complex, and likely different for each variable, even though phase-phase interactions in discharge clearly show the signature of those identified in precipitation.

## 4.3   Conclusion

Those findings allow for a better identification of climate deterministic processes controlling hydroclimate variability, notably using cross-scale analysis, which could help identifying more robust climate drivers.

For instance, it is important to discriminate pure climate influence from climate-land processes interactions. This has large implications for seamless hydrological predictions based on climate information, as only some parts of the climate signals are transferred to discharge systems. Thus, causal cross-scale relationship could be used to inform and improve existing seasonal to multi-year seamless forecasting for hydrological variability, including extremes (e.g. flood and drought). Preliminary work in this direction were recently

proposed by Jajcay et al. (2018), who developed a composite binning method enabling to forecast a particular time series based on conditional phase of another. Similarly, it would be of crucial importance to determine whether hydrological models, which are commonly used in climate-impact assessments, are reproducing the filtering-processes induced by the catchment properties, and identify those (Ducharne et al., 2020). Long term hydroclimate variability only represents a fraction of the total variability, however, strong interactions

between high and low frequency variations have been highlighted. Those interactions are both spatial and temporal (Feliks et al., 2016). Owing to the recent addition of long term, high spatial resolution hydroclimate data sets (e.g. Fyre reconstructions, Devers et al. (2020, 2021)), it is now possible to apply the clustering and cross-scale analyses to better characterize the effects that long term hydroclimate variability (e.g. multi-decadal) has on smaller time scales. The methodology presented in this work can enable deeper analyses

than those based on correlations, which may overlook some important hydroclimate processes.

**Code/Data Availability**

**Data**

Safran Precipitation and Temperature dataset must be obtained from www.meteofrance.fr
Discharge data is available from http://www.hydro.eaufrance.fr


**Code**

The code used for this study is available at https://github.com/ManuelFossa/Hess-2021-81

Pyclits python code used for Cross-scale interactions is available at https://github.com/jajcayn/pyclits

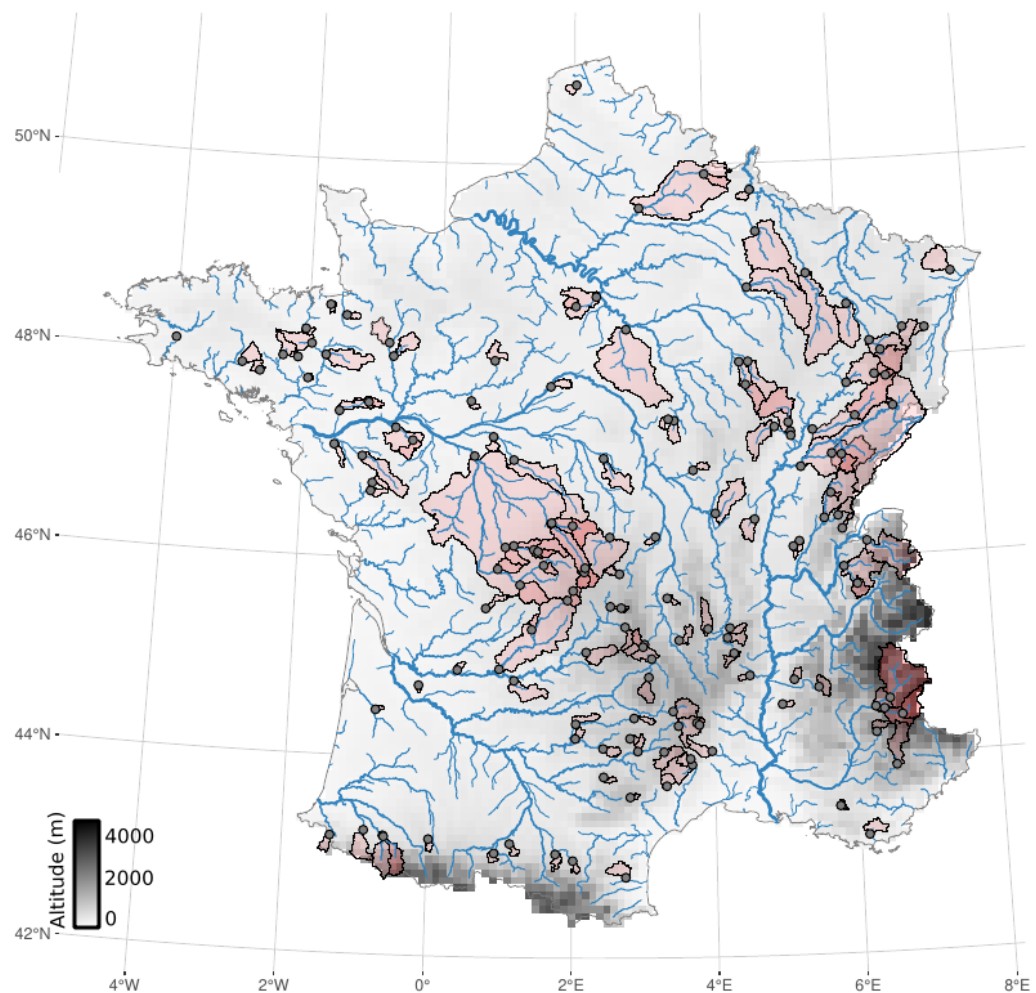

**Figure 1.** Location of stream gauges (grey dots), corresponding watersheds (pale red, Brigode et al. (2020)), hydrographic network (blue lines, Pella et al. (2012)), and orography in Safran dataset (grey scale, Vidal et al. (2010)

.

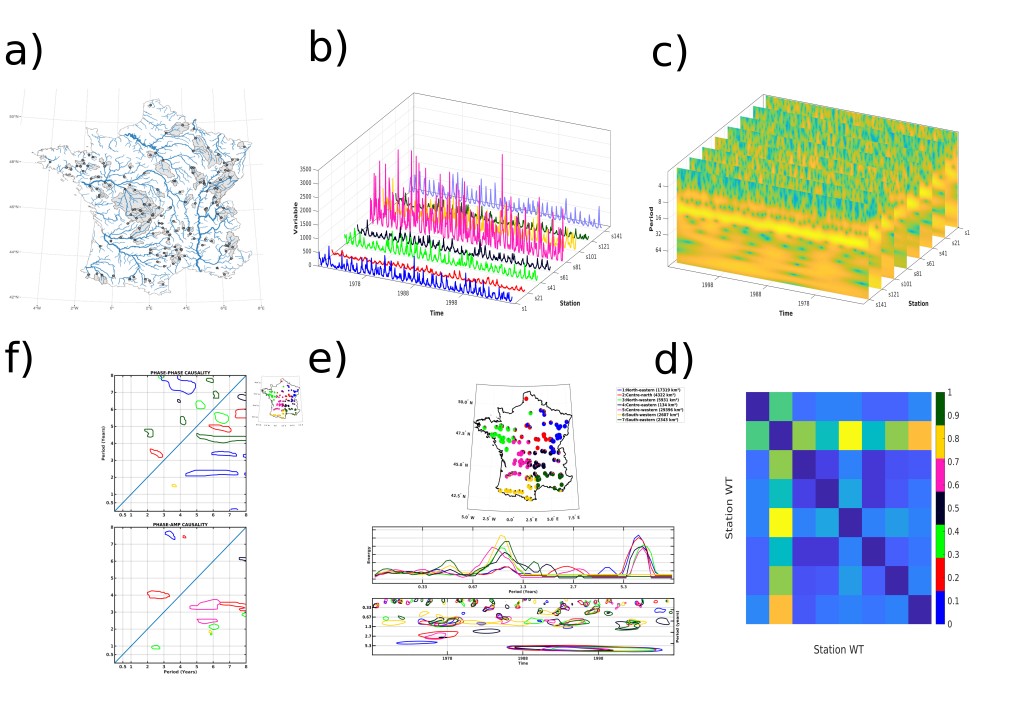

**Figure 2.** The workflow of this study is repeated for precipitation, temperature and discharge data sets. (**a**) 152 near-natural watersheds are selected. (**b**) Each watershed is represented by a monthly time series from 1968 to 2008, discharge is measured at a gauging station, precipitation and temperature are averaged over the watershed. (**c**) The continuous wavelet transform of each time series is computed, representing the time scale dependent, non-stationary variability of each watershed. (**d**) The similarity between all 152 continuous wavelet transforms is computed and represented as distance matrix. (**e**) *top)* Similar watersheds are grouped together into regions of homogeneous variability, using a fuzzy clustering algorithm; *middle)* For each region, the relative importance of each time scale dependent variability is represented with a global wavelet spectrum, and all regions global wavelet spectra are superimposed; *bottom)* The continuous wavelet spectra of regions are superimposed. For clarity, only time scale and time locations with the most significant variability are shown (coloured circles). (**f**) For each region, cross time scales interactions are computed. *top)* phase-phase interactions identify any time scale's phase that conditions another time scale's phase.; *bottom)* The phase-amplitude interactions characterize any time scale's phase that conditions the amplitude of another time scale.

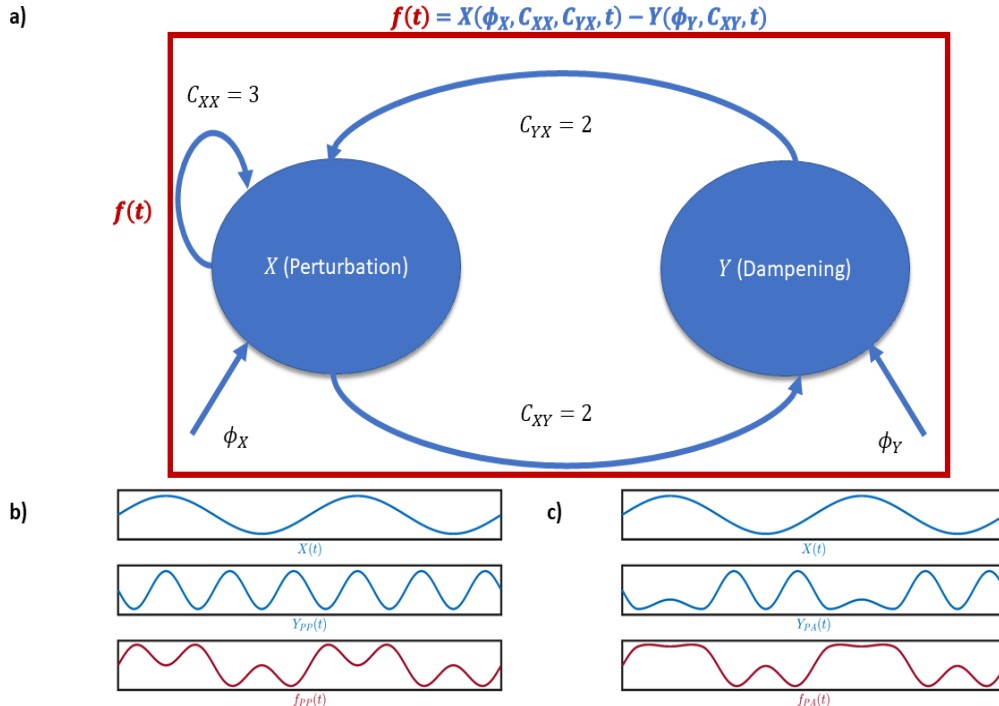

**Figure 3.** A system with directional cross-scale interactions. **a)** A variable $f(t)$ made of two components $X$ and $Y$, connected through $C_{XY}$ and $C_{YX}$ in a perturbation-dampening scheme so that $f(t) = X(t) - Y(t)$. Both $X$ and $Y$ receive inputs $\phi_X$ and $\phi_Y$, respectively. $C_{XX}$ allows $X$ to grow first. Depending on both inputs and connections, some phase-phase or phase-amplitude interactions between $X$ and $Y$ can occur. **b)** an example of a phase-phase interaction, with every fourth ridge of $Y_{PP}$ coinciding with a ridge of $X$, with $f_{PP}(t) = X(t) - Y_{PP}(t)$ (top,middle and bottom panels, respectively). **c)** an example of phase-amplitude interaction. $X$ and $Y_{PA}$ only interact when $X$ reaches a ridge, in which case $Y_{PA}$ amplitude if lowered, yielding $f_{PA}(t)$ (Top, middle and bottom panels, respectively). (adapted from Onslow et al. (2014))

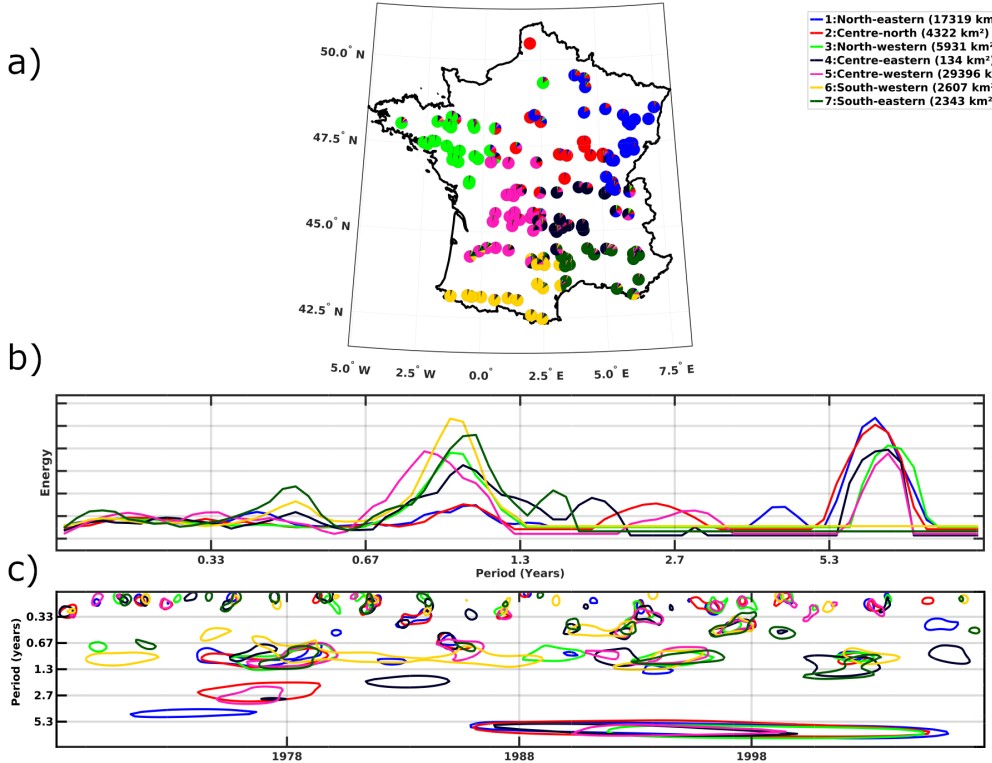

**Figure 4.** Clustering of precipitation time-frequency variability in France. (**a**) Classification map of the watersheds. Pie charts slices show the three highest probability memberships. Pie charts denote fuzzy clustering memberships. (**b**) Global wavelet spectra of homogeneous regions.(**c**) Wavelet spectra of homogeneous regions. For clarity, only time scales and time locations 95% statistically significant and with the largest variability are shown (coloured circles).

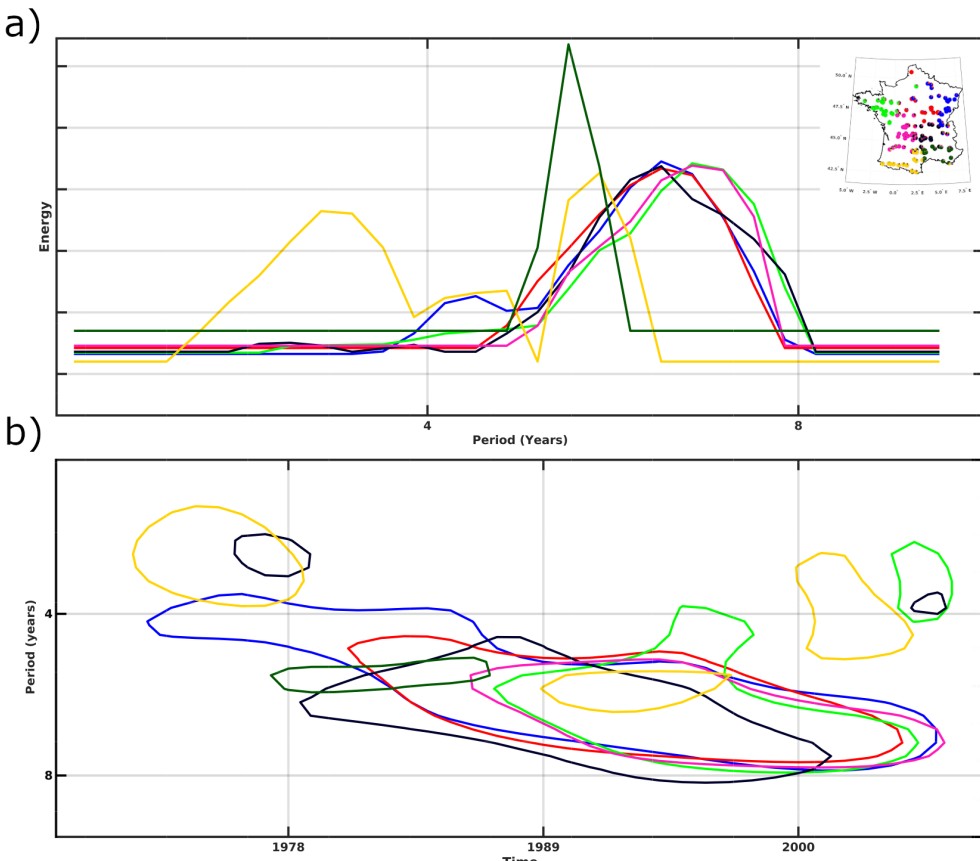

**Figure 5.** Inter-annual precipitation time-frequency variability in France. (**a**)Global wavelet spectra of homogeneous regions. (**b**) Wavelet spectra of homogeneous regions. For clarity, only time scales and time locations with the 95% statistically significant and largest variability are shown (coloured circles).

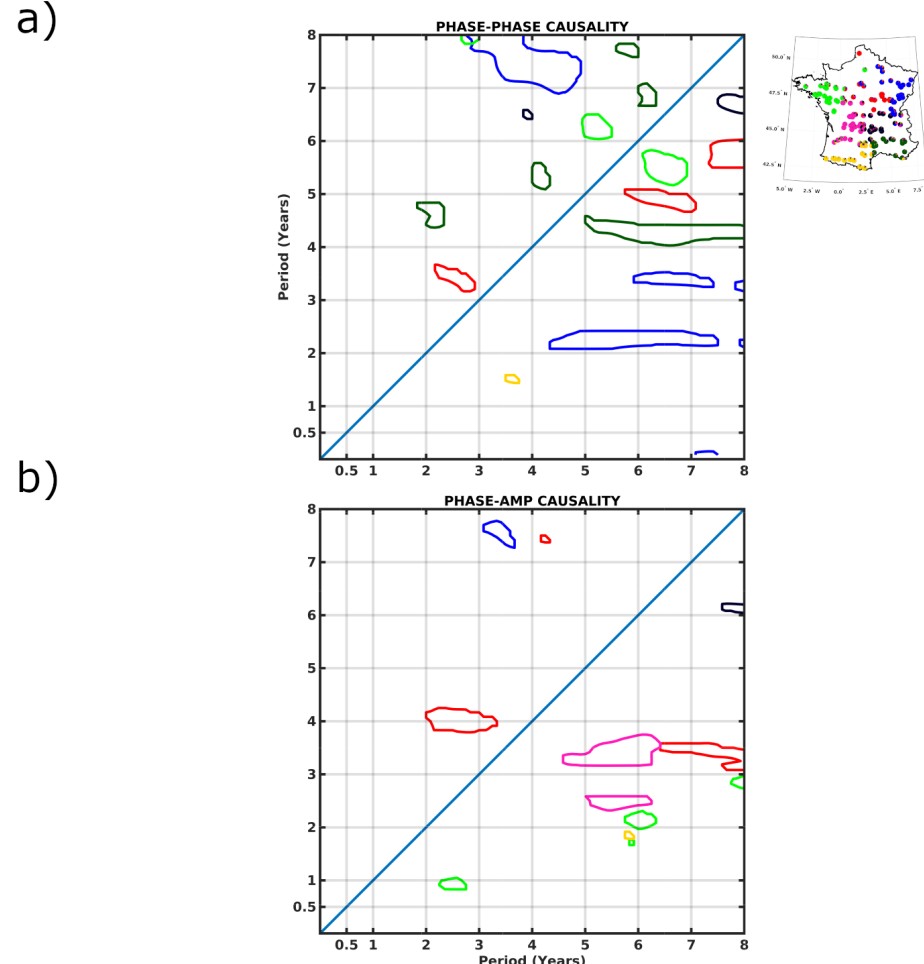

**Figure 6.** Precipitation cross-scale interactions (95% significance level).The driving time scale is on the horizontal axis, the driven on the vertical axis (*i.e.* the time scale $x$ phase has a causal relationship with the phase/amplitude of the driven time scale $y$). Lower(upper) half of the graph, below (above) the diagonal, show time scales acting on smaller (larger) time scales. (**a**) Phase-phase causality. (**b**) Phase-amplitude causality.

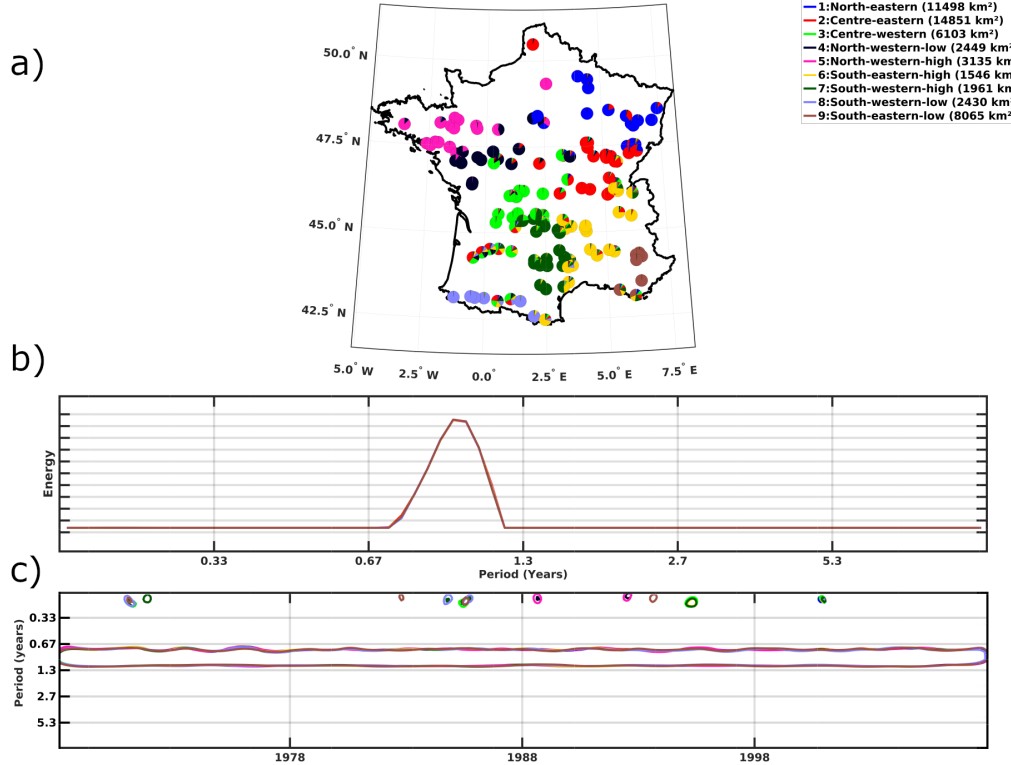

**Figure 7.** Clustering of temperature time-frequency variability in France. (**a**) Classification map of the watersheds. Pie charts slices show the three highest probability memberships (**b**) Global wavelet spectra of regions. (**c**) Wavelet spectra of homogeneous regions. For clarity, only time scales and time locations with the 95% statistically significant and largest variability are shown (coloured circles).

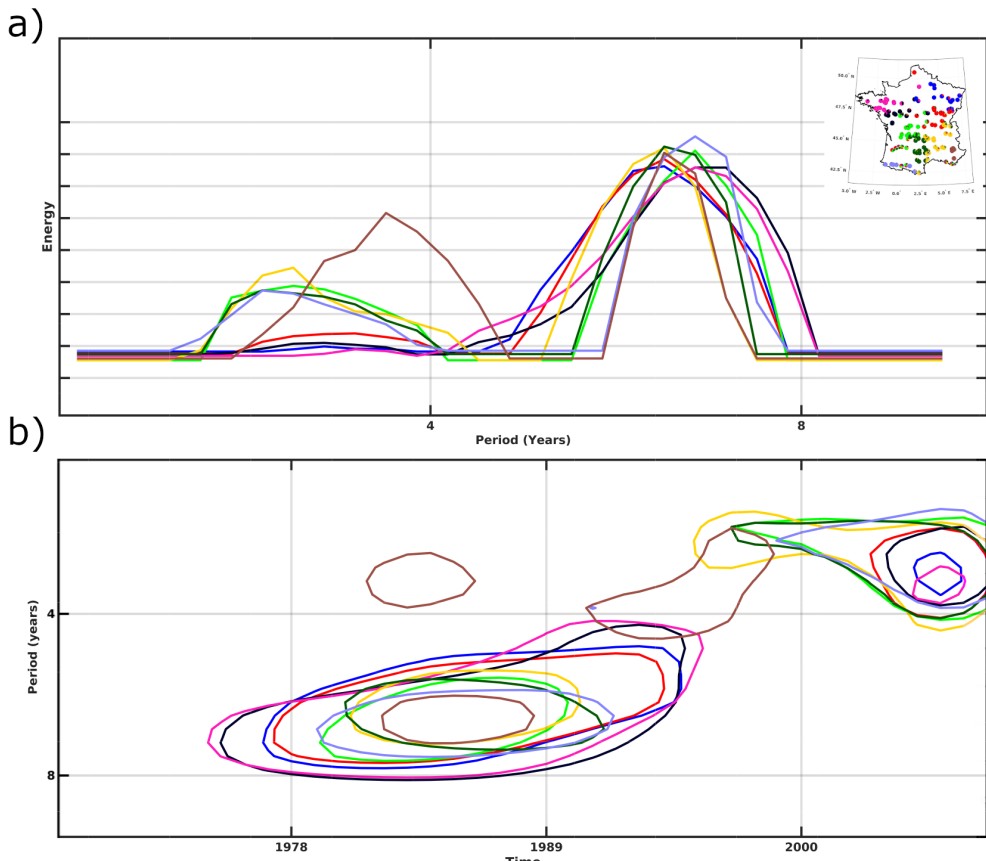

**Figure 8.** Inter-annual temperature time-frequency variability in France. (**a**) Global wavelet spectra of homogeneous regions. (**b**) Wavelet spectra of homogeneous regions. For clarity, only time scales and time locations 95% statistically significant and with the largest variability are shown (coloured circles).

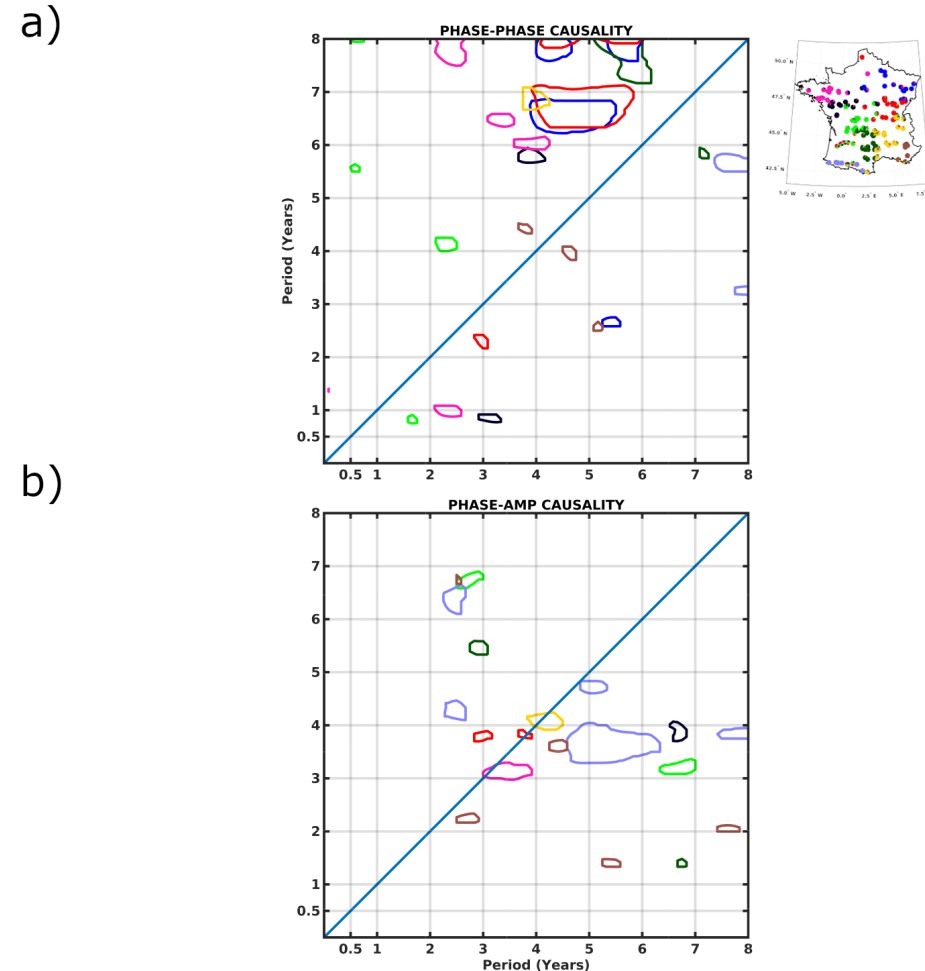

**Figure 9.** Temperature cross-scale interactions (95% significance level).The driving time scale is on the horizontal axis, the driven on the vertical axis (*i.e.* the time scale $x$ phase has a causal relationship with the phase/amplitude of the driven time scale $y$). Lower(upper) half of the graph, below (above) the diagonal, show time scales acting on smaller (larger) time scales. (**a**) Phase-phase causality. (**b**) Phase-amplitude causality

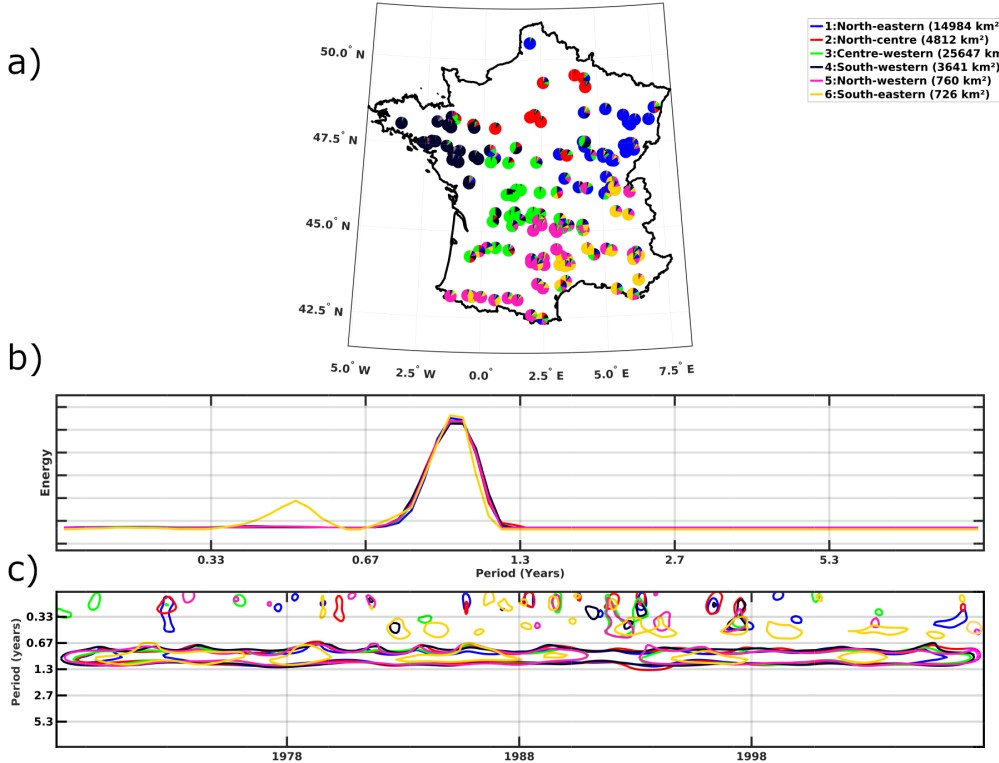

**Figure 10.** Clustering of discharge time-frequency variability in France. (**a**) Classification map of the watersheds. Pie charts slices show the three highest probability memberships (**b**) Global wavelet spectra of homogeneous regions. (**c**) Wavelet spectra of homogeneous regions. For clarity, only time scales and time locations 95% statistically significant and with the largest variability are shown (coloured circles).

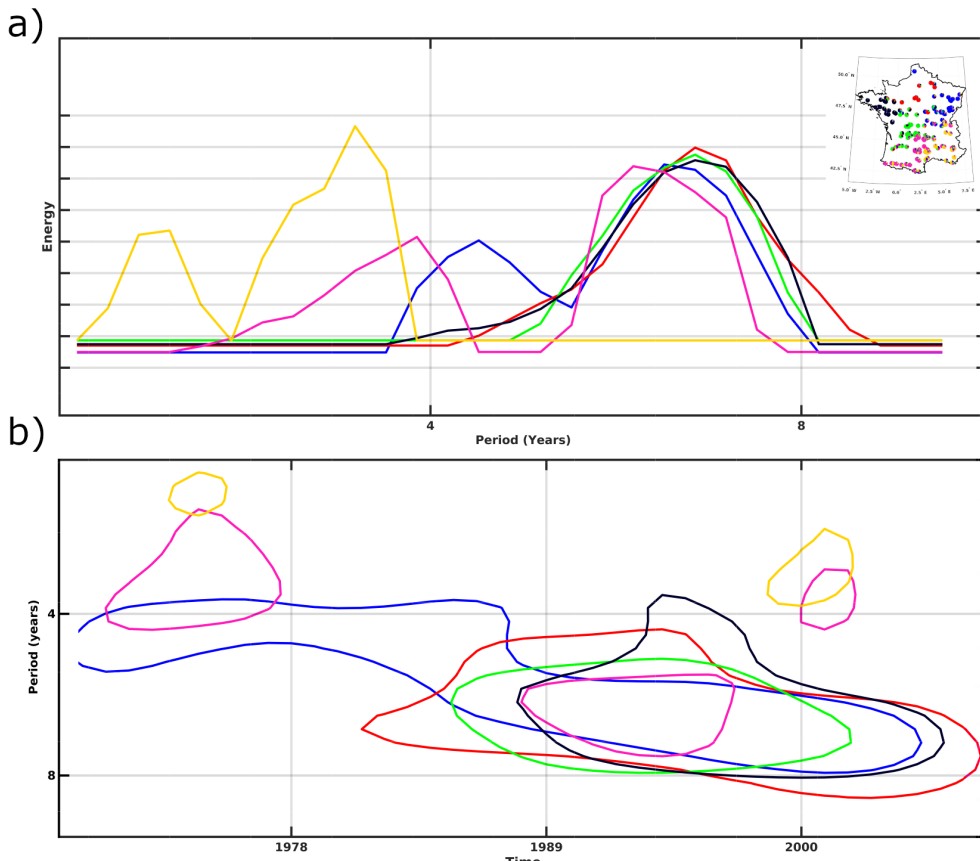

**Figure 11.** Inter-annual discharge time-frequency variability in France. (**a**) Global wavelet spectra of homogeneous regions. (**b**) Wavelet spectra of homogeneous regions. For clarity, only time scales and time locations 95% statistically significant and with the largest variability are shown (coloured circles).

a)

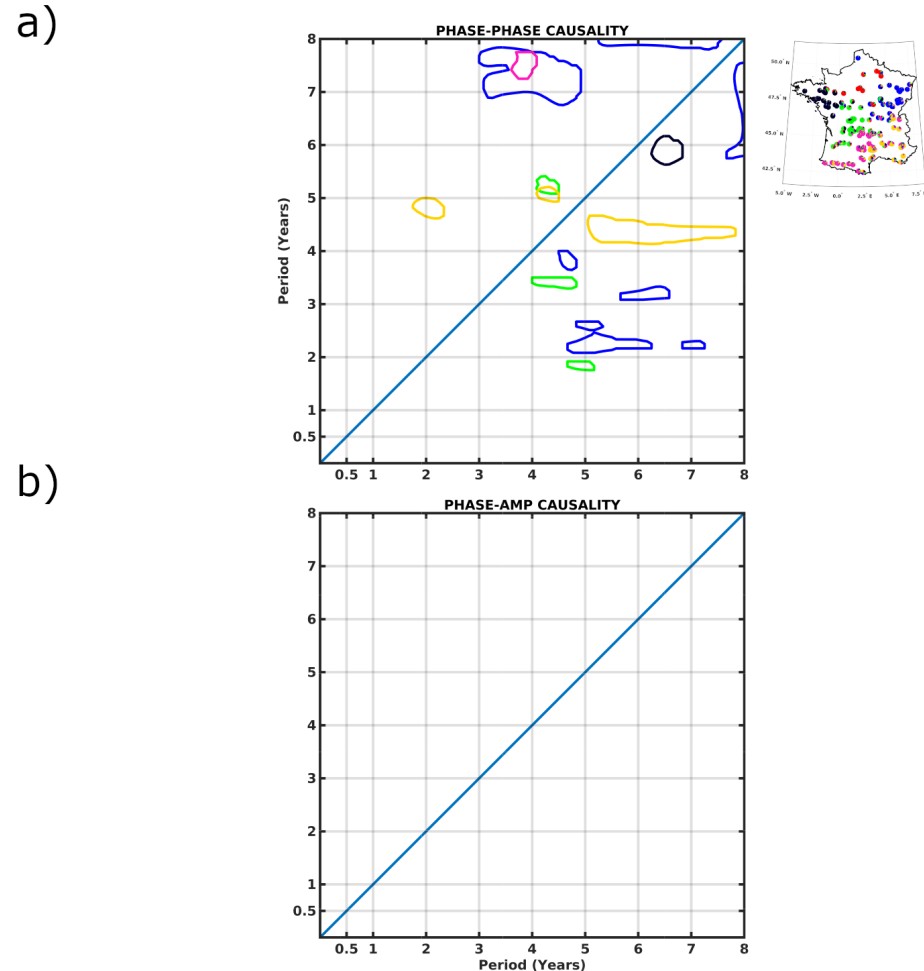

b)

**Figure 12.** Discharge cross-scale interactions (95% significance level).The driving time scale is on the horizontal axis, the driven on the vertical axis (*i.e.* the time scale $x$ phase has a causal relationship with the phase/amplitude of the driven time scale $y$). Lower(upper) half of the graph, below (above) the diagonal, show time scales acting on smaller (larger) time scales. (**a**) Phase-phase causality. (**b**) Phase-amplitude causality

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
