# Peer review of "SPATIO-TEMPORAL AND CROSS-SCALE INTERACTIONS IN HYDROCLIMATE VARIABILITY: A CASE-STUDY IN FRANCE"

_Hydrology and Earth System Sciences, 2021_

## Referee Comment (RC2)

[referee-annotated manuscript omitted]

---

## Author Comment (AC1)

**SPATIO-TEMPORAL AND CROSS-SCALE INTERACTIONS IN HYDROCLIMATE VARIABILITY: A CASE-STUDY IN FRANCE**

Manuel Fossa[1], Bastien Dieppois[2], Nicolas Massei[1], Matthieu Fournier[1], Benoit Laignel[1], and Jean-Philippe Vidal[3]

[1] Normandie Univ, UNIROUEN, UNICAEN, CNRS, M2C, 76000 Rouen, France
[2] Centre for Agroecology, Water and Resilience (CAWR), Coventry University, Coventry, UK
[3] INRAE, UR Riverly, 5 rue de la Doua, CS 20244, 69625 Villeurbanne Cedex, France

**Correspondence:** Manuel Fossa (manuel.fossa1@univ-rouen.fr)

**1 Response to RC1**

*The authors present a wavelet-based framework for understanding the variability in precipitation, temperature and streamflow in France. Especailly interesting is the cross-scale analysis. However, the discussion of these results were not well done. The authors have only reported the results. More discussion on these trowing light on the physical intuition and learning derived*

5 *is needed. It was also not clear if the cross scale implied across spatial scales ... in which case, one driving others is confusing as the manifestation of a variable, while having spatial variability, need not necessarily always have a causal connection. In the case of streaflow, this is clear due to the network nature of the flow. If the cross-scale is across wavelet scales, in that, for each time series, the dependence is between low-frequency vs. high frequency, then how does this dependence manifest in space needs to be disscussed.*

10

We thank Referee 1 for his comments. Section 2.2.4, Line 99 specifies that Cross-scale interactions refer to phase-phase and phase-amplitude couplings between time scale of a given time series, so no spatial information is accounted in the methodology used. Additionally, cross scale interactions are estimated on each homogeneous region as identified by the clustering algorithm so spatial information within each region is also lost. The spatial dependence of those cross-scale interactions is discussed

15 by comparing the characteristics of those interactions for each region. A much more complex test would be to test the cross scale interactions between any pair of stations, but that would be prohibitive in terms of computational cost. This is however an interesting topic, that could be explored in a future work. We also took into consideration the lack of physical insight and have provided additional explanation of our results, especially for cross-scale interactions. We started from the characteristics highlighted in Figure 3, and linked them climate and watershed characteristics known to show variability at the time scales

20 detected in the cross-scale interactions.

---

## Author Comment (AC2)

**SPATIO-TEMPORAL AND CROSS-SCALE INTERACTIONS IN HYDROCLIMATE VARIABILITY: A CASE-STUDY IN FRANCE**

Manuel Fossa1, Bastien Dieppois2, Nicolas Massei1, Matthieu Fournier1, Benoit Laignel1, and Jean-Philippe Vidal3

1 Normandie Univ, UNIROUEN, UNICAEN, CNRS, M2C, 76000 Rouen, France

2 Centre for Agroecology, Water and Resilience (CAWR), Coventry University, Coventry, UK

3 INRAE, UR Riverly, 5 rue de la Doua, CS 20244, 69625 Villeurbanne Cedex, France

Correspondence: Manuel Fossa (manuel.fossa1@univ-rouen.fr)

**1 Response to RC2**

The authors want to thank RC2 for his comments, literature suggestions and very helpful detailed review. Responses to each point in RC2's additional pdf are provided below.

5 *Line36: non-stationary in the long-term (I imagine this is what the authors mean here)*

We use the strong stationarity definition that is, the joint probability distribution is not time dependent. Thus, a non-stationary signal is a signal with a joint probability distribution that changes with time. It is not tied to any long term period, or scale, and can be present at any time.

10

Line 76: Maybe it is worth to add a paragraph explaining how the authors dealt with changes in river management: were there any dam construction, or management through direct water uptake, during the chosen period? This element could artificially modify the results.

15 The dataset used is a near-natural watersheds compilation by Giuntoli et al 2013, thus anthropogenic modifications were discarded as much as possible. We think discussing the methods used to determine the near naturalness of the watersheds is beyond the scope of this article, however, we mention the procedure used by Giuntoli et al. 2013, in the revised paper.

Line 182. What do you mean with annual variability? Seasonal? Mixing seasonal and inter-annual variability is actually interesting for precipitation, in areas where the seasonal cycle is very low. In these regions, the seasonal cycle does not overwhelm other long-term variances, and this should be highly emphasized. In the NE and CE cluster the seasonal variability (1-year-period) is much lower than the 6-8-year variability. Instead, in areas featured by Mediterranean climate (SW and SE), almost only the seasonal cycle is detectable. These considerations deserve a wider description. Also, fig. 4c needs more explanation on its physical meaning. It is not clear to me how significant can be, for instance, the NW cluster (light green) for the 8-year period. The 8-year cycle seems to be observed over a period shorter than 16 years (two cycles).

In the article we use the terms "intra-seasonal", "annual" and "inter-annual" for corresponding wavelet periods of less than one year, one year, 2-4 and 5-8 years respectively. The terminology comes from the wavelet analaysis framework, onto which cross-scale interactions analysis is also based. We have added a paragraph explaining the terminology in the methods section (Wavelet analysis sub-section).

Line 185. Figure 5 is almost neglected, while it is important because now the seasonal cycle and the interannual and multiannual scales are not mixed. Here too I have doubt on the statistical significance, though. The yellow cluster has its 6-year-cycle only between 1989 and 1997? More physical insight is needed, I am still puzzled by these results.

35

25

30

Both in the fuzzy clustering sub-section, and below figures showing wavelet transform, we omitted the precision that the clustering was made on the entire wavelet spectrum of each time series, and not just on the significant areas and that it is only for showing the differences between each region that we only show those significant areas in the Figures. The clustering is made on the entire wavelet spectrum because due to the nature of the statistical tests, significant areas in two regions may be

- 40 at a very similar frequency-time location but just slight shifted one versus the other. Also, the total area of significance on a wavelet spectrum is usually small (*i.e.* there are only few "circles") and thus those slight differences would bias the clustering, by giving too much weight in the estimation of the distance between two wavelet spectra to those slight differences. The areas outside of significant areas, even if they are true noise, are usually similar between wavelet spectra as well, so they tend to have a very small bias. It is thus more consistent to cluster entire wavelet spectra than just their significant areas. Note that in wavelet
- 45 analysis, significant areas are always areas of the highest variability, so those areas do show the most striking features of each region, but we may have mentioned such details in the paper. We revised both the wavelet analysis methodology sub-section and mention that in the figures, we show only significant areas for clarity reasons, that is, to be able to graphically compare the regions. We also added more discussion on Figure 5 (and subsequent inter-annual wavelet analysis figures).
- 50 *Line 222. yes, but on seasonal scale (e.g. Ardilouze et al., 2021), but on longer time-scales soil moisture hardly play a role (even if feedback mechanisms can prolong the effect of soil moisture on precipitation Bellucci et al., 2021)*

We thank RC2 for providing us those very useful references, which we used in the framework of a deeper discussion, in conjunction with a deeper discussion of the feedback dynamics that are necessarily present in case of cross scales interactions to offer some physical insight on the role of both climate and soil state forcing on the identified interactions.

Line 268: citation needed after soil moisture (Miralles et al., 2012; Materia et al., 2021)

The references provided have been of great help to support the revised discussion. We added discussion about cross-scale 60 interactions differences between southern and northern regions thanks to the provided literature.

Figure 1:A legend for the topography is needed.

A legend as well as a color bar was added to Figure 1.

**65**

Figure 2: should be the one that explains the entire methodology, and in fact it does not. What do (a), (b), (c), (d) refer to? The figure does not show different panels to justify the letters in the caption. I strongly suggest a figure that makes the methodology, which is rather complex, clear to the readers.

70 We agree with RC2 that Figure 2 was not informative enough. Figure 2 has been completely redrawn, and the workflow detailed.